# Psychiatric Comorbidity in Mexican Adolescents with a Diagnosis of Eating Disorders Its Relationship with the Body Mass Index

**DOI:** 10.3390/ijerph18083900

**Published:** 2021-04-08

**Authors:** David Ruiz-Ramos, José Jaime Martínez-Magaña, Ana Rosa García, Isela Esther Juarez-Rojop, Thelma Beatriz Gonzalez-Castro, Carlos Alfonso Tovilla-Zarate, Emmanuel Sarmiento, María Lilia López-Narvaez, Humberto Nicolini, Alma Delia Genis-Mendoza

**Affiliations:** 1División Académica de Ciencias de la Salud, Universidad Juárez Autónoma de Tabasco, Villahermosa 86100, Mexico; daruiz_914@hotmail.com (D.R.-R.); jimy.10.06@gmail.com (J.J.M.-M.); iselajuarezrojop@hotmail.com (I.E.J.-R.); thelma.glez.castro@gmail.com (T.B.G.-C.); 2Instituto Nacional de Medicina Genómica (INMEGEN), Ciudad de México 14610, Mexico; hnicolini@inmegen.gob.mx; 3Hospital Psiquiátrico Infantil Juan N, Navarro, Secretaría de Salud, Ciudad de México 14080, Mexico; anarosagarciab@gmail.com (A.R.G.); emmanuelsarmientoh@hotmail.com (E.S.); 4División Académica Multidisciplinaria de Comalcalco, Universidad Juárez Autónoma de Tabasco, Comalcalco 86100, Mexico; alfonso_tovillaz@yahoo.com.mx; 5Hospital General de Yajalón “Dr. Manuel Velazco Siles”, Secretaría de Salud, Yajalón 29930, Mexico; dralilialonar@yahoo.com.mx

**Keywords:** psychiatric comorbidity, Mexican population, eating disorders

## Abstract

The prevalence of comorbid psychiatric disorders among patients with eating disorders (ED) is higher than the general population. Individuals diagnosed with eating disorders have changes in their body mass index which could promote severe metabolic disruptions. This study aimed (1) to report the prevalence of comorbid psychiatric disorders among a Mexican adolescent sample diagnosed with eating disorders, (2) to compare our results with the prevalence of psychiatric disorders reported from a national survey of mental health of adolescents, (3) to compare the presence of psychiatric comorbidities between ED diagnoses, and (4) to explore the relationship of these comorbidities with the body mass index. In the study, we included 187 Mexican adolescents diagnosed with eating disorders. The psychiatric comorbidities were evaluated using the Mini International Neuropsychiatric Interview for children/adolescents, and a revised questionnaire on eating and weight patterns. We found that 89% of the Mexican adolescents diagnosed with ED had another psychiatric comorbidity. Major depressive disorder (52.40%) and suicide risk (40%) were the most prevalent comorbidities. Attention and deficit hyperactivity disorder (ADHD) prevalence was different between ED diagnosis, and adolescents with binge-eating disorder and ADHD had the higher body mass index. Our results showed that in this sample of Mexican adolescents, the presence of comorbidities could impact body mass index. This emphasizes the importance that clinicians take into consideration the presence of psychiatric comorbidities to achieve an integrative treatment for adolescents diagnosed with ED.

## 1. Introduction

Eating disorders (ED) are multifactorial psychiatric disorders (anorexia nervosa, bulimia nervosa and, binge eating disorder), characterized by alterations in eating patterns and body weight perceptions [1,2]. ED is a public health problem. The prevalence of eating disorders among the female adolescent population is 0.3–2.05% and 0.1–0.8% for the male population [3]. In the Mexican population, the prevalence of anorexia nervosa (AN) in the adolescent population 0.3%, and 0.9% for bulimia nervosa (BN) [4].

One focus of attention has been centered on the increase of mortality among patients with ED in comparison to the general population [5]. In this sense, an association between ED with the manifestation of other psychiatric disorders (comorbidity) has been reported to be a possible risk factor for the increase in the mortality rates in these patients [6]. Different psychiatric disorders as comorbidities (alcohol use disorder, substance use disorder, depression, and personality disorders) have been associated with ED [5]. In the Mexican population, drug and alcohol abuse, and borderline personality disorder have been reported to be the most prevalent comorbidities in adults diagnosed with ED [7].

The main physical impact of ED is seen in body mass index (BMI). Individuals diagnosed with AN generally have a reduced BMI; in contrast, individuals diagnosed with BN or BED tend to have a higher BMI [8]. BMI is an extensively used indicator of several health issues, like obesity, diabetes, and hypertension [9]. Additionally, neurobiological circuits of phenotypical traits (impulsivity) have been postulated to regulate food intake and body mass index (BMI), assembling a complex relation between psychiatric disorders and metabolism. In this sense, patients with obesity could demonstrate disruptive feeding behavior that leads to physical and functional impairment in diseases like bipolar disorder [10], schizophrenia [11], and attention-deficit/hyperactivity disorder (ADHD) [12].

However, in the Mexican population and adolescents, there is limited information about the prevalence of comorbid psychiatric conditions among adolescent patients with ED, most studies had been performed on adolescents that migrated to or were born in the United States [13,14]. In Mexico, some studies had been performed on substance use disorder, but only on female adolescents [15]. Also, the national surveys of psychiatric disorders have not reported the prevalence of ED, but the reported prevalence could be compared to the reported prevalence in the general adolescent sample [16]. The aim of this study was (1) to report preliminary results on the prevalence of comorbid psychiatric disorders among a sample of Mexican adolescents diagnosed with ED, (2) to compare our results with the prevalence reported from a national survey of mental on adolescents, (3) to compare the presence of psychiatric comorbidities between ED diagnoses. As well as (4) to explore if the comorbidities had a relation with body mass index.

## 2. Materials and Methods

### 2.1. Study Population

The present study is a cross-sectional study with convenience sampling. The recruitment of the sample was in the areas of continuous admission and external consultation of the Dr. Juan N. Navarro Children’s Psychiatric Hospital in Mexico City, from 2017 to 2020. Children and adolescents were invited to participate in the study by the psychiatry team, explaining that not participating in the study will not affect the treatment. The psychiatric hospital is one of the national reference institutions for child and adolescent psychiatric disorders. As inclusion criteria, the patients were recruited from different services, like the emergency room and outpatient clinic. A psychiatrist specializing in eating-disorders evaluated and diagnosed the patients. The diagnosis was based on the Diagnostic and Statistical Manual of Mental Disorders 5 (DSM-5) criteria [17]. In addition, the Spanish versions of the Mini International Neuropsychiatric Interview for Children and Adolescent (MINI-kid) [18,19] and Questionnaire on Eating and Weight Pattern-Revised (QEWP-R) [20], were applied to explore other psychiatric comorbidities and eating-related patterns. Only adolescents with ages between 12 to 17 years were included. Also, to be included, parents and grandparents had to be Mexican. The subjects were excluded from the study if either (1) missing data was found on the questionnaires or (2) the parents’ participants (or tutors) withdrew their consent for participation in the study.

### 2.2. Ethical Statement

All patients included were given verbal and written information related to the research objectives and procedures. The patients read and signed an informed assent, and informed consent was given by parents or tutors. The subjects were informed of anonymity, and they did not receive economical remuneration. This study was in accordance with the principles of the Helsinki declaration in 1975 and compliance with the code of ethics of the world medical association. The protocol was approved by the ethics and investigation committees of the Instituto Nacional de Medicina Genómica (INMEGEN) (approbation number: 06/2018/I) and the Dr. Juan N. Navarro Children´s Psychiatric Hospital.

### 2.3. Measurements

#### 2.3.1. Sociodemographic and Anthropometric Measurements

Age, gender, scholarship, and parents/grandparents’ country of birth were collected using a structured questionnaire of all individuals. The anthropometric measurements collected included weight and height as previously reported [21]; the body mass index (BMI) was determined according to the obesity task force criteria. One of the limitations of the use of BMI on children and adolescents is that this parameter could be influenced by the development of the child. In order to make the comparison between the BMI, we transformed this parameter into z-score (z-BMI) values (s.d., standard deviation), and age and gender-specific BMI percentiles (BMI percentiles), as previously reported [22,23]. We defined underweight, overweight, and obese based on the 2007 WHO growth chart reference for school-age children and adolescents: underweight as z-score of BMI < −2 s.d., overweight as BMI z-score > 1 s.d. and <= 2 s.d., and obese as BMI z-score > 2 s.d. [22,23].

#### 2.3.2. Clinical Measurements

##### Mini International Neuropsychiatric Interview for Children and Adolescents (MINI-Kid)

The MINI-kid is a short, standardized, structured, diagnostic interview, designed for diagnostic criteria of the DSM-IV and ICD-10 for psychiatric disorders. MINI-kid disorders have shown test-retest reliability and validity comparable to other standardized diagnostic interviews [24]. In the Mexican population, it has been validated [25].

##### Questionnaire on Eating and Weight Pattern-Revised (QEWP-R)

The QEWP-R was developed by Spitzer [26] to define binge-eating disorder. This version was updated to align more closely with BED criteria [20]. It has been validated in different languages like Portuguese [27] and Spanish [28]. The QEWP-R has been reported to be a useful screening tool for BED.

### 2.4. Statistical Analysis

In general, we reported the mean and standard deviations (s.d.) for continuous variables; and frequencies and percentages for categorical variables. Statistical analysis was performed with packages available in R software version 3.5.1 (https://CRAN.R-project.org, accessed 30 December 2020) [29]. The significance level was set at *p* < 0.05. Next, we described the statistical contrast performed.

#### 2.4.1. Comparison with Reference Sample

In order to explore whether the prevalence of the comorbidities found in our sample was higher or lower than those reported for the adolescent Mexican population, we compared our calculated prevalence with the reported in a national epidemiological survey of adolescents performed in Mexico [16]. The comparison data was obtained from Benjet C., et al. [16]; this study is an epidemiologic survey of psychiatric disorders among the Mexican adolescent population with a sample of over 3000 subjects, it was carried out at the Ramon de la Fuente National Institute of Psychiatry (further citation as reference sample). A chi-squared test was calculated to compare the prevalence of psychiatric comorbidities in our study population with the reported prevalence of psychiatric disorders among the adolescent Mexican population.

#### 2.4.2. Comparison between ED Diagnoses

The prevalence of comorbidities between the three main ED diagnoses (AN, BN, and BED) was performed with a chi-squared test, using a 2 (presence or absence of psychiatric disorder) × 3 (AN, BN, and BED) contingency table.

#### 2.4.3. Comparison of BMI between ED Diagnosis and Comorbidities

Once we established the psychiatric-comorbidities prevalence differences between ED diagnoses, we explored the effect that the presence of those disorders could have on body mass index (in this analysis we use the z-score of BMI). A Shapiro-Wilkins test was applied to the z-score BMI to explore normality distribution. A Student’s *t*-test with Welch adjustment was performed to explore differences between groups of individuals with these disorders. Next, a chi-squared test was performed to determine a significant difference between groups with the BMI stratification (normal-weight, overweight, and obese).

## 3. Results

In the present study, 187 adolescents diagnosed with eating disorders were included. There was a higher number of females in the sample compared to males [female 77% (*n* = 144); male 23% (*n* = 43)]. The mean age was 14.08 (s.d. ± 1.7) years old. Of the total of patients, 32 adolescents patients (*n* = 32, 17.11%) were diagnosed with anorexia nervosa, 104 patients (*n* = 104, 55.61%) with bulimia nervosa, and finally, 51 patients (*n* = 51, 27.27%) with binge eating disorder.

The BMI-for-age percentile was 74.26 ± 25.3, meanwhile the mean z-score for BMI was 0.8 ± 1. Based in growth charts, six patients (*n* = 6, 3.20%) were underweight; 93 patients (*n* = 93, 49.73%) had a normal-weight; 34 patients (*n* = 34, 18.18%) were overweight and 54 patients (*n* = 54, 28.87%) presented obesity. From the sample of patients with BN, almost 40% (*n* = 45) of the sample were overweight (*n* = 21, 20.19%) or obese (*n* = 24, 23.08%). Whereas for the patients with BED, more than 80% (*n* = 41) of the individuals were overweight (*n* = 12, 23.53%) or obese (*n* = 29, 56.86%).

### 3.1. Major Depressive Disorder Was the Most Frequent Psychiatric Comorbidity

We observed that 89.30% (*n* = 167) of the adolescents with ED had another psychiatric comorbidity. Patients diagnosed with BN or BED had a higher prevalence of any psychiatric comorbidity (BN 90.38%, *n* = 94; and BED 94.11%, *n* = 48). Moreover, patients with AN had a lower rate with 78.12% (*n* = 25) of the prevalence of another psychiatric comorbidity.

The most prevalent comorbidity in individuals with ED were disorders related to mood alterations. Major depressive disorder was the most prevalent comorbidity with 52.40% (*n* = 98) of the sample. Noteworthy, over 40% (*n* = 76) of the adolescents with ED presented suicide risk. In lesser prevalence was the dysthymic disorder with 22.45% (*n* = 42). In relation to comorbidities related to anxiety disorders, generalized anxiety disorder was the most prevalent with 17.64% (*n* = 33). However, a higher prevalence than anxiety-related disorders was seen related to developmental disorders, 20.85% (*n* = 39) of the patients with ED had attention-deficit/hyperactivity disorder (ADHD). Finally, the less prevalent comorbidities in the patients with ED were oppositional defiant disorder (13.90%, *n* = 2), psychotic disorders (6.95%, *n* = 13) and substance use disorder (5.34%, *n* = 10).

### 3.2. Prevalence of Psychiatric Disorders among Patients with EDs Is Higher Than the Reference Sample

We compared the prevalence of comorbid psychiatric disorders in patients with ED with the previously reported prevalence in the reference sample [18]. Of the 20 disorders compared between the ED sample and the reference sample, 11 disorders showed statistically significant differences. Almost all the disorders had a higher prevalence in the ED sample compared to the reference, except for specific phobia, where we found a reduction of 10% in prevalence in ED. Major depressive disorder was the disorder that showed the higher difference between the ED sample and the reference, with almost an increase of 50%.

Next, the disorders that showed more differences in prevalence were suicide risk (an increase of 31.89%), dysthymic disorder (an increase of 21.96%), attention-deficit/hyperactivity disorder (an increase of 19.26%), and generalized anxiety disorder (an increase of 17.15%). We did not compare some comorbidities (adjustment disorder, psychotic disorder, TIC disorder, and obsessive-compulsive disorder) for lack of information in the reference sample. The prevalence of comorbidities in our study sample and statistical results of the comparisons with the references are shown in Table 1.

### 3.3. Attention-Deficit/Hyperactivity Disorder Prevalence Was Different between the Diagnosis of ED and It Is Associated with an Increased BMI in BED

Further, we compared the differences in prevalence between the groups of ED diagnosis (i.e., differences between AN, BN, and BED). ADHD and adjustment disorder were the only comorbidities with a significant difference between the groups (Table 2 and Figure 1). Patients diagnosed with BED (*n* = 16, 31.37%) had the highest prevalence of ADHD, compared to BN (*n* = 20, 19.23%) and AN (*n* = 3, 9.38%). In adjustment disorder, also individuals diagnosed with BED had a higher prevalence (*n* = 5, 9.80%), indicating in AN there was not an individual diagnosed with adjustment disorder. The comparison of prevalence stratified by ED diagnosis and statistical results of the comparisons are shown in Table 2.

In order to explore the effect that these comorbidities (i.e., ED and ADHD) could have on the BMI in adolescents, we compared the differences of the BMI between individuals with ED-ADHD and those that did not have ADHD. For this analysis, we did not include individuals diagnosed with AN, because of the reduced sample size. Also, we did not perform a comparison with adjustment disorder for the same reason. We made comparisons of the z-score of BMI for the patients with BN and BED. The individuals with BED and ADHD had a higher prevalence of obesity, compared to individuals with BED without ADHD (*p* = 0.0481) (Table 3). Even still, individuals with BED and ADHD had higher z-BMI than those individuals with BN and ADHD (*p* = 0.0009). (Figure 2).

## 4. Discussion

In the present work, we performed an analysis of the prevalence of psychiatric comorbidities in adolescent individuals diagnosed with eating disorders in a Mexican sample. In our sample, we found a high prevalence of mood disorders among patients with eating disorders. Our results are similar to previous studies, an increase in mood disorders in patients with ED has been reported [30,31]. Patients diagnosed with BED were found to have higher levels of comorbidity when compared with patients with AN or BN [31]. In this sense, the criteria diagnosis for BED has been recently developed and the presence of comorbidities on this diagnostic group has been scarcely explored. The increased prevalence of other psychiatric comorbidities in patients with ED could result in a negative impact on illness course and possible treatment outcome [32]. Noteworthy, almost 40% of the individuals diagnosed with BN or BED presented suicide risk and the prevalence is even 30% higher than the reference sample.

The estimates of suicide risk in our population are in accordance with other reports, where an estimation with a range of 40% to 60% has been reported [33,34,35]. Suicide risk could be a marker of other suicide conducts, like suicide attempts or suicide completion [36]. This higher prevalence of suicide risk in the population with ED has to be a point to be considered in further analysis, due to the high risk in mortality associated with suicide, which could even be increased in adolescents diagnosed with ED. In this sense, a follow-up of the effect that these comorbidities could have on the development of the disorder has to be explored but this analysis is outside of the aims of the present study. Nevertheless, the high presence of suicide risk and depression could be a delicate subject to be considered during the treatment of individuals diagnosed with ED in the adolescent Mexican population. Mainly, because a high percentage of individuals with suicide ideation do not receive treatment [37].

Our findings showed that almost 50% of the individuals diagnosed with ED, presented as overweight or obese, mainly patients with BN and BED. There is a close relationship between alteration of eating-patterns and changes in the modulation of BMI [38,39]. Even when the mechanisms underlying this relationship are stilled under investigation, some hypotheses have been generated [40]. In this sense, one hypothesis suggested that alterations in body weight (i.e., overweight or obesity) are processes influenced by a dysregulation in neurobiological circuits that control some behaviors, like impulsivity and reward sensitivity [41]. These alterations in the regulation of reward-impulsivity neurobiological circuits could lead to uncontrolled eating, those are the main points of discussion around this hypothesis. The regulations of these impulsivity-reward brain circuits have been mapped to brain areas like the ventral tegmental area, lateral hypothalamus, prefrontal cortex, amygdala, and hippocampus [42,43]. Some findings had also reported alterations in these impulsivity-reward brain circuits in patients with ED, which could be a link between these complex traits [44,45,46]. Our results could be reinforcing this hypothesis (i.e., alteration of impulsivity-reward brain circuits in ED diagnosed patients).

A relationship was found between ADHD, BED, and obesity. Similar results were found in childhood and adults, from other populations [47]. About ADHD and eating behavior, some results point-out a genetic risk factor that could be modulating this relationship, the fat mass and obesity-associated gene (FTO). FTO is known as a post-genome-wide association study (GWAS) gene, this appellation was established because the association of FTO and BMI was discovered by a mean of a molecular technique known as GWAS [48]. In recent studies, genetic variants of FTO, principally the tag-SNP rs9939609, have been established as a genetic risk factor for ED and ADHD [49,50,51]. Another FTO gene polymorphism (rs1421085, in high correlation with rs9939609) has been associated with alterations on brain areas principally regulating impulsivity [52]. Also, FTO genetic variant rs9939609 has been associated with alcohol dependence [53]. The possible relationship of FTO in ED-ADHD and impulsivity-reward alterations remain undercover but it is an interesting one to explore further [54]. The higher BMI found on individuals with BED and ADHD is a point of clinical relevance, mainly on the effect that this increase could have in metabolic disorders, like metabolic syndrome onset [55,56,57], where this comorbidity became important not only from a mental health perspective but also from a physical perspective. Our study had some limitations, the patients recruited in our work were considerably small in comparison with the reference sample. The reduced sample size, impact the possible comparison that we could make, like in the comparisons of BMI and ADHD, where we could not perform the comparison in the individuals diagnosed with AN. Another effect was seen with the adjustment disorder and BMI, and the reduced sample size did not allow us to perform statistical contrasts. Also, the extrapolation to the general population of our findings was not possible. However, a larger sample could asseverate the differences found in our preliminary results.

## 5. Conclusions

In conclusion, we observed a high prevalence of depressive disorders, and anxiety- and stressor-related disorders among adolescent patients with eating disorders. Patients with eating disorders had a higher prevalence of comorbidities in comparison with the reference sample. Finally, patients with BED had a higher prevalence of ADHD and this comorbidity could be impacting directly on BMI. This emphasizes the importance that clinicians take into consideration the presence of psychiatric comorbidities to achieve an integrative treatment for patients with ED.

## Figures and Tables

**Figure 1 ijerph-18-03900-f001:**
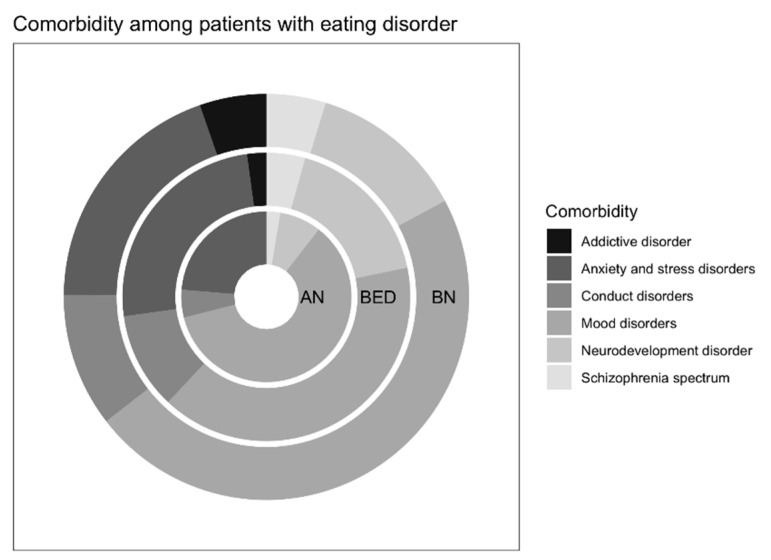
Comorbidity among patients with an eating disorder. Abbreviations: AN, anorexia nervosa; BN, bulimia nervosa; BED, binge eating disorder.

**Figure 2 ijerph-18-03900-f002:**
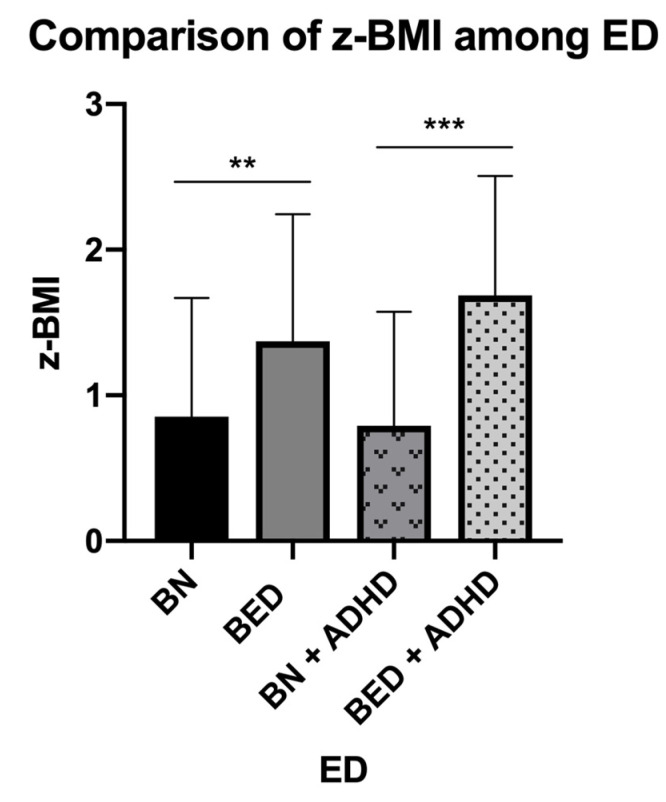
Comparison of z-BMI values among patients with ED and ED-ADHD. Abbreviations: ED, eating disorders; AN, anorexia nervosa; BN, bulimia nervosa; BED, binge eating disorder; ADHD, attention-deficit/hyperactivity disorder. Notes: ** indicates *p* = 0.017 of a Student’s *t*-test; *** indicates *p* = 0.009 of a Student’s *t*-test.

**Table 1 ijerph-18-03900-t001:** Comparison of psychiatric comorbidity prevalence in patients with eating disorders and reported prevalence in the Mexican adolescent population.

Psychiatric Comorbidity	EDs n (%)	Reported Prevalence n (%) ^a^	X^2^	*p*-Value
Major depressive disorder	98 (52.40)	144 (4.79)	**562.85**	**<0.001**
Dysthymic disorder	42 (22.45)	15 (0.49)	**484.09**	**<0.001**
Hypomanic disorder	3 (1.60)	75 (2.49)	0.58	0.620
Suicide risk	76 (40.64)	214 (8.75)	**176.84**	**<0.001**
Panic disorder	9 (4.81)	48 (1.59)	**10.37**	**0.004**
Agoraphobia	9 (4.81)	108 (3.59)	0.43	0.500
Separation anxiety disorder	2 (1.06)	78 (2.59)	1.67	0.320
Social phobia	12 (6.41)	336 (11.18)	3.63	0.056
Specific phobia	19 (10.16)	628 (20.89)	**11.90**	**<0.001**
Generalized anxiety disorder	33 (17.64)	15 (0.49)	349.50	**<0.001**
Obsessive-compulsive disorder	3 (1.60)	-	-	-
Post-traumatic stress disorder	6 (3.20)	30 (0.99)	**7.71**	**0.010**
Adjustment disorder	7 (3.74)	-	-	-
Alcohol use disorder	3 (1.60)	96 (3.19)	1.48	0.200
Substance use disorder (non-alcohol)	10 (5.34)	39 (1.29)	**19.10**	**<0.001**
Conduct disorder	11 (5.88)	90 (2.99)	**3.89**	**0.040**
Oppositional defiant disorder	26 (13.90)	150 (4.99)	**25.15**	**< 0.001**
TIC disorder	4 (2.13)	-	-	-
Attention–deficit/hyperactivity disorder	39 (20.85) ^b^	48 (1.59)	**239.05**	**<0.001**
Psychotic disorder	13 (6.95)	-	-	-

**Abbreviations**: ED, eating disorders; X^2^, Square Chi. **Notes**: Bold indicates significant statistical *p*-value < 0.05; Fisher´s exact test was applied when values were <5. ^a^ Benjet, C., et al., Youth mental health in a populous city of the developing world: results from the Mexican Adolescent Mental Health Survey. *J. Child Psychol. Psychiatry*
**2009**, *50*, 386–395. ^b^ Analysis between groups X^2^ = 6.13; *p* = 0.04.

**Table 2 ijerph-18-03900-t002:** Comparison of psychiatric comorbidities in Mexican patients with eating disorders.

Psychiatric Comorbidity	AN n = 32 n (%)	BN n = 104 n (%)	BED n = 51 n (%)	X^2^	*p*-Value
Mood disorders	23 (71.88)	80 (76.92)	37 (72.55)	0.53	0.767
Major depressive disorder	14 (43.75)	55 (52.88)	29 (56.86)	1.37	0.502
Dysthymic disorder	8 (25.00)	21 (20.19)	13 (25.49)	0.69	0.706
Hypomanic disorder	0	3 (2.88)	0	2.43	0.745
Suicide risk	9 (28.13)	47 (45.19)	20 (39.22)	3.01	0.221
Anxiety and stressor related disorders	9 (28.23)	33 (31.73)	23 (45.10)	3.44	0.178
Panic disorder	1 (3.13)	4 (3.85)	4 (7.84)	1.43	0.578
Agoraphobia	1 (3.13)	4 (3.85)	4 (7.84)	1.43	0.578
Separation anxiety disorder	1 (3.13)	1 (0.96)	0	1.84	0.387
Social phobia	2 (6.25)	6 (5.77)	4 (7.84)	0.24	0.917
Specific phobia	3 (9.38)	13 (12.50)	3 (5.88)	1.66	0.452
Generalized anxiety disorder	4 (12.50)	17 (16.35)	12 (23.53)	1.91	0.411
Obsessive-compulsive disorder	1 (3.13)	2 (1.92)	0	1.36	0.575
Post-traumatic stress disorder	1 (3.13)	2 (1.92)	3 (5.88)	1.72	0.381
Adjustment disorder	**0**	**2** (1.92)	**5** (9.80)	**7.39**	**0.042**
Substance-related and addictive disorders	0	9 (8.65)	2 (3.92)	3.79	0.178
Alcohol use disorder	0	3 (2.88)	0	2.43	0.745
Substance use disorder (non-alcohol)	0	8 (7.69)	2 (3.92)	3.14	0.232
Disruptive, Impulse-control and conduct disorders	2 (6.25)	18 (17.31)	10 (19.61)	2.88	0.228
Conduct disorder	1 (3.13)	7 (6.73)	3 (5.88)	0.57	0.913
Oppositional defiant disorder	2 (6.25)	16 (15.38)	8 (15.69)	1.89	0.454
Neurodevelopment disorders	**3 (9.38)**	**21 (20.19)**	**16 (31.37)**	**5.85**	**0.058**
TIC disorder	1 (3.13)	1 (0.96)	2 (3.92)	1.61	0.284
Attention–Deficit/hyperactivity disorder	**3 (9.38)**	**20 (19.23)**	**16 (31.37)**	**6.13**	**0.046**
Psychotic disorder	1 (3.13)	8 (7.69)	4 (7.84)	0.87	0.791

Abbreviations: AN, anorexia nervosa; BN, bulimia nervosa; BED, binge eating disorder; X^2^, Square Chi Notes: Bold indicates significant statistical *p*-value < 0.05; Fisher´s exact test was applied when values were <5. The psychotic disorder was considered as schizophrenia spectrum.

**Table 3 ijerph-18-03900-t003:** Comparisons between BMI diagnoses in patients with BN and BED with the presence or absence of ADHD in the Mexican adolescent population.

	Bulimia Nervosa			Binge Eating Disorder		
BMI Diagnoses	No ADHD (*n* = 83)	ADHD (*n* = 20)	X^2^; *p*-Value	No ADHD (*n* = 35)	ADHD (*n* = 16)	X^2^; *p*-Value
Normal weight	45 (54.22)	13 (65.00)	1.07; 0.586	8 (22.86)	2 (12.50)	6.00; 0.048
Overweight	17 (20.48)	4 (20.00)		11 (31.43)	1 (6.25)	
Obesity	21 (25.30)	3 (15.00)		16 (45.71)	13 (81.25)	

Abbreviations: ADHD (attention-deficit/hyperactivity disorder), X^2^ (Chi square). Note: Bold indicates *p* < 0.05.

## Data Availability

The data presented in this study are available on request from the corresponding author. The data are not publicly available due to the confidentiality of the data and the signing of the informed consent.

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
