# Peer review of "Psychiatric Comorbidity in Mexican Adolescents with a Diagnosis of Eating Disorders Its Relationship with the Body Mass Index"

_ijerph, 2021, doi:10.3390/ijerph18083900_

Round 1
Reviewer 1 Report
This study aims to estimate the prevalence of comorbid psychiatric disorders among a Mexican adolescent sample (12-17 years old) diagnosed with eating disorders. It assesses 187 Mexican adolescents and compares its data with a previous study with a sample of 3000 adolescents. Although the admirable effort of the authors to explore and enrich the paper findings. I think that the authors could take a step further in making clear the novelty/innovation of this study, the relevance of the statistical analysis, as organizing study methodology and results in a more intelligible and explanatory way.
Editing by a native speaker is required throughout the manuscript. Finally, I believe that this data has a great potential for more robust and innovative statistical analysis.
General Comments
Abstract
- Please avoid the use of acronyms in the abstract (e.g. QEWP-R). The sample size should be included in this section.
Introduction
- The pertinence and innovation of this study are not clear. It will be helpful to interpret data in order to make a stronger case for the pertinence of this work. The introduction could provide a more strong background.
- “Additionally, neurobiological circuits of phenotypical traits (impulsivity) have been postulated to regulate food intake and body mass index (BMI), assembling a complex relation between psychiatric disorders and metabolism. In this sense, patients with obesity could demonstrate disruptive feeding behavior that leads to physical and functional impairment in diseases like bipolar disorder [7], schizophrenia [8] and attention-deficit/hyperactivity disorder (ADHD) [9].” This idea about obesity seems disconnected from the previous paragraph and from the study aims.
- “there is limited information about the prevalence of comorbid psychiatric conditions among adolescent patients with ED.” Please support this sentence with the appropriate references. It will be helpful to know what information exists about the prevalence of comorbid psychiatric conditions among Mexican and non-Mexican adolescent patients with ED.
- Study aims are not fully congruent with the study results (e.g. differences between two studies are explored in the results section but not in the study aims).
Methods
- There is missing procedural information that makes this study difficult to assess. For example, this is a cross-sectional study? What information were potential participants given about the study? What was the response rate? How specifically were participants recruited? Data collection is still ongoing?
- The statistical analysis section is limited. I would suggest a more detailed section explaining how you performed the statistical analysis. It will also be important to know if the variables under study were normally distributed. The information and rationale about the comparison data (Benjet et al.) should be described in the introduction.
- X2 – Please include the name of the statistical test rather than just the symbol.
- A measures section is missing. In this section authors cloud include a more detailed description of the sociodemographic, clinical and anthropometric measures applied in this study.
- It is important to clarify throughout the paper if you are referring to the biological sex of participants or to their self-attributed/chosen “gender”.
Results
- A sample size of 187 adolescents for an epidemiological/prevalence study is an important limitation. In spite of the relevance of the data collected, generalizations to the Mexican population based on a sample size of 187 adolescents are not appropriated.
- Please try to include the sample size together with the corresponding percentages (e.g. 17.64% (n = x).
- In Table 2, the authors also reported differences in the adjustment disorder. What was the authors' rationale to only highlight differences in Attention-deficit/hyperactivity disorder?
- “Attention-deficit/hyperactivity disorder prevalence was different between the diagnosis of ED and could modulate BMI”. Please reformulate this subtitle. It implies a mediation analysis that was not performed.
- Since BMI z score is a numeric variable did the authors consider to performed other statistical analyses rather than chi-square? Such as for example an independent sample t-test? Nevertheless, I noticed that did not exist enough variation in the data and sample size to detect differences with the AN subsample. In table 3, a rationale should be included to explain why these patients were excluded from further analysis.
- Regarding figure 2 what statistical analyses were performed to explore differences?
- Rather than focus on prevalence, additional statistical analysis could be performed to enrich the paper such as correlations or a path analysis to understand more clearly the relationships between the variables under study for each ED group.
Discussion
- It will be valuable to include a more detailed description of the possible relevance and clinical implications of this study for the Mexican population.
- Study limitations should be described in more detail.
Thank you for the opportunity to revise your valuable work. I hope that my comments help to improve your paper.
Author Response
Reviewer #1.
Comments and Suggestions for Authors
This study aims to estimate the prevalence of comorbid psychiatric disorders among a Mexican adolescent sample (12-17 years old) diagnosed with eating disorders. It assesses 187 Mexican adolescents and compares its data with a previous study with a sample of 3000 adolescents. Although the admirable effort of the authors to explore and enrich the paper findings. I think that the authors could take a step further in making clear the novelty/innovation of this study, the relevance of the statistical analysis, as organizing study methodology and results in a more intelligible and explanatory way.
Editing by a native speaker is required throughout the manuscript. Finally, I believe that this data has a great potential for more robust and innovative statistical analysis.
General Comments
Abstract
Comment. Please avoid the use of acronyms in the abstract (e.g. QEWP-R). The sample size should be included in this section.
Response. We included the sample size in the abstract and avoid the use of abbreviations.
- In the study we included 187 Mexican adolescents diagnosed with eating disorders.
- Mini International Neuropsychiatric for children/adolescents, and Questionnaire on eating and weight patterns – revised.
Introduction
Comment. The pertinence and innovation of this study are not clear. It will be helpful to interpret data in order to make a stronger case for the pertinence of this work. The introduction could provide a more strong background.
- “Additionally, neurobiological circuits of phenotypical traits (impulsivity) have been postulated to regulate food intake and body mass index (BMI), assembling a complex relation between psychiatric disorders and metabolism. In this sense, patients with obesity could demonstrate disruptive feeding behavior that leads to physical and functional impairment in diseases like bipolar disorder [7], schizophrenia [8] and attention-deficit/hyperactivity disorder (ADHD) [9].” This idea about obesity seems disconnected from the previous paragraph and from the study aims.
- “there is limited information about the prevalence of comorbid psychiatric conditions among adolescent patients with ED.” Please support this sentence with the appropriate references. It will be helpful to know what information exists about the prevalence of comorbid psychiatric conditions among Mexican and non-Mexican adolescent patients with ED.
Response. We updated the introduction, to be more congruent.
- The main physical impact of ED is seen in Body Mass Index (BMI). Individuals diagnosed with AN mainly have a reduced BMI; in contrast, individuals diagnosed with BN or BED tend to have a higher BMI (doi: 10.1002/erv.2166). BMI is an extensively used indicator of several health issues, like obesity, diabetes, and hypertension (doi: 10.1097/NT.0000000000000092). Additionally, neurobiological circuits of phenotypical traits (impulsivity) have been postulated to regulate food intake and body mass index (BMI), assembling a complex relation between psychiatric disorders and metabolism. In this sense, patients with obesity could demonstrate disruptive feeding behavior that leads to physical and functional impairment in diseases like bipolar disorder [7], schizophrenia [8] and attention-deficit/hyperactivity disorder (ADHD) [9].
- However, in the Mexican population and adolescents, there is limited information about the prevalence of comorbid psychiatric conditions among adolescent patients with ED, most studies had been performed on adolescents that migrated or were born in the United States (doi: 10.1001/archgenpsychiatry.2011.22;doi: 10.1002/eat.20406).
Comment. Study aims are not fully congruent with the study results (e.g. differences between two studies are explored in the results section but not in the study aims).
Response. Thanks, we modified the aims.
- The aim of this study was 1) to report as preliminary results the prevalence of comorbid psychiatric disorders among a sample of Mexican adolescents diagnosed with ED, 2) to compare our results with the prevalence reported from a national survey of mental on adolescents, 3) to compare the presence of psychiatric comorbidities between ED diagnoses. As well as 4) to explore if the comorbidities had a relation with Body Mass Index.
Methods
Comment. There is missing procedural information that makes this study difficult to assess. For example, this is a cross-sectional study? What information were potential participants given about the study? What was the response rate? How specifically were participants recruited? Data collection is still ongoing?
Response. We added the missing information.
- The present study is a cross-sectional study with convenience sampling. The recruitment of the sample was in the areas of continuous admission and external consultation of the children´s psychiatric hospital “Dr. Juan N. Navarro” in Mexico City, from 2017 to 2020. Children and adolescents were invited to participate in the study by the psychiatry, explaining that not participating in the study will not affect the treatment.
Comment. The statistical analysis section is limited. I would suggest a more detailed section explaining how you performed the statistical analysis. It will also be important to know if the variables under study were normally distributed. The information and rationale about the comparison data (Benjet et al.) should be described in the introduction.
Response. we updated the statistical description.
- 4 Statistical analysis
In general, we reported the mean and standard deviations (s.d) for continuous variables; and frequencies and percentages for categorical variables. Statistical analysis was performed with packages available in R software version 3.5.1 (https://CRAN.R-project.org) [19]. The significance level was set at p < 0.05. Next, we described the statistical contrast performed.
- 4.1 Comparison with reference sample
In order to explore whether if the prevalence of the comorbidities found in our sample were higher or lower than those reported for the adolescent Mexican population, we compared our calculated prevalence with the reported in a national epidemiological survey of adolescents performed in Mexico [18]. The comparison data was obtained from Benjet C., et al [18]; this study is an epidemiologic survey of psychiatric disorders among Mexican adolescent population with a sample of over 3 thousand subjects, it was carried out at the National Institute of Psychiatry “Ramon de la Fuente” (further citation as reference sample). A chi-squared test was calculated to compare the prevalence of psychiatric comorbidities in our study population in with the reported prevalence of psychiatric disorders among the adolescent Mexican population.
- 4.2. Comparison between ED diagnoses
The prevalence of the comorbidities between the three main ED diagnoses (AN, BN, and BED) was performed with a chi-squared test, by mean of a 2 (presence or absence of psychiatric disorder) x 3 (AN, BN, and BED) contingency table.
- 4.3. Comparison of BMI between ED diagnosis and comorbidities
Once we established the psychiatric-comorbidities prevalence differences between ED diagnoses, we explored the effect that the presence of those disorders could have on Body Mass Index (in this analysis we the z-score of BMI). Shapiro-Wilkins test was applied to z-score BMI to explore normality distribution. A T Student test with Welch adjustment was performed to explore differences between groups of individuals with those disorders. Next, a chi-squared test was performed to determine a significant difference between groups with the BMI stratification (normal-weight, overweight, and obesity).
Comment. X2 – Please include the name of the statistical test rather than just the symbol.
Response. we include the statistical test name.
Comment. A measures section is missing. In this section authors cloud include a more detailed description of the sociodemographic, clinical and anthropometric measures applied in this study.
Response. we added a section of measurements.
- 3 Measurements
2.3.1 Sociodemographic and Anthropometric measurements
Age, gender, scholarship, and parents/grandparents’ country of birth were collected using a structured questionnaire of all individuals. The anthropometric measurements collected included weight and height as previously reported [15]; the body mass index (BMI) was determined according to the obesity task force criteria. One of the limitations of the use of BMI on children and adolescents is that this parameter could be influenced by the development of the child. In order to make the comparison between the BMI, we transformed this parameter into z-score (z-BMI) values (s.d, standard deviation), and age and gender-specific BMI percentiles (BMI percentiles), as previously reported [16, 17]. We defined underweight, overweight, and obesity based on the 2007 WHO growth chart reference for school-age children and adolescents: underweight as z-score of BMI < -2 s.d, overweight as BMI z-score > 1 s.d and <= 2 s.d, and obese as BMI z-score > 2 s.d [16, 17].
- 3.2 Clinical measurements
Mini International Neuropsychiatric Interview for children and adolescents (MINI-kid).
The MINI-kid is a short standardized structured diagnostic interview, design for diagnostic criteria of the DSM-IV and ICD-10 for psychiatric disorders. Mini-kid disorders have shown test-retest reliability and validity comparable to other standardized diagnostic interviews (https://doi.org/10.1186/s12888-019-2121-8). In the Mexican population, it has been validated (De la Peña OF, Esquivel AG, Pérez GAJ, Palacios CL. Validación Concurrente para Trastornos Externalizados del MINI-Kid y la Entrevista Semiestructurada para Adolescentes. Rev Chil Psiquiatr Neurol Infanc Adolesc 2009;20(1):8-12).
- Questionnaire on Eating and Weight Pattern-Revised (QEWP-R).
The QEWP-R was developed by Spitzer (https://doi.org/10.1002/1098-108X(199204)11:3<191::AID-EAT2260110302>3.0.CO;2-S) to define binge-eating disorder. This version was updated to align more closely with BED criteria (doi:10.1002/eat.22372). It has been validated in different languages like Portuguese (https://doi.org/10.1590/S1516-44462005000400012) and Spanish (doi: 10.1590/S0212-16112012000200031). The QEWP-R has been reported to be a useful screening tool for BED.
Comment. It is important to clarify throughout the paper if you are referring to the biological sex of participants or to their self-attributed/chosen “gender”.
Response. we updated the term to gender.
Results
Comment. A sample size of 187 adolescents for an epidemiological/prevalence study is an important limitation. In spite of the relevance of the data collected, generalizations to the Mexican population based on a sample size of 187 adolescents are not appropriated.
Response. we included a description of this limitation.
- The reduced sample size, impact in the possible comparison that we could made, like in the comparisons of BMI and ADHD, were we could not performed the comparison in the individuals diagnosed with AN. Another effect was seen in the effect of the adjustment disorder and BMI, were the reduced sample size did not allow to performed statistical contrasts. Also, the extrapolation to the general population of our findings could not be possible.
Comment. Please try to include the sample size together with the corresponding percentages (e.g. 17.64% (n = x).
Response. we updated the sample sizes.
Comment. In Table 2, the authors also reported differences in the adjustment disorder. What was the authors' rationale to only highlight differences in Attention-deficit/hyperactivity disorder?
Response. we updated the result section with this result. ADHD and adjustment disorder were the only comorbidities with significate difference between the groups.
Comment. “Attention-deficit/hyperactivity disorder prevalence was different between the diagnosis of ED and could modulate BMI”. Please reformulate this subtitle. It implies a mediation analysis that was not performed.
Response. we change the subtitle.
Comment. Since BMI z score is a numeric variable did the authors consider to performed other statistical analyses rather than chi-square? Such as for example an independent sample t-test? Nevertheless, I noticed that did not exist enough variation in the data and sample size to detect differences with the AN subsample. In table 3, a rationale should be included to explain why these patients were excluded from further analysis.
Response. For this analysis we did not included the individuals diagnosed with AN, because of the reduced samples size. Also, we did not performed comparison with adjustment disorder for the same reasons
Comment. Regarding figure 2 what statistical analyses were performed to explore differences?
Response. we update the statistical section, and also the note of the Figure.
Comment. Rather than focus on prevalence, additional statistical analysis could be performed to enrich the paper such as correlations or a path analysis to understand more clearly the relationships between the variables under study for each ED group.
Response. we believe the correlations or path analysis, could be interesting and more sophisticate analysis to performed in the future, with a higher sample size.
Discussion
Comment. It will be valuable to include a more detailed description of the possible relevance and clinical implications of this study for the Mexican population.
Response. we added a section focusing principally in suicide risk.
- Nevertheless, the high presence of suicide risk and depression could be a delicate subject to be considered during the treatment of individuals diagnosed with ED in the adolescent Mexican population. Mainly because a high percentage of individuals with suicide ideation does not received treatment (doi: 10.1192/bjp.bp.110.084129).
Comment. Study limitations should be described in more detail.
Response. we expanded the limitations section.
- The reduced sample size, impact in the possible comparison that we could made, like in the comparisons of BMI and ADHD, were we could not performed the comparison in the individuals diagnosed with AN. Another effect was seen in the effect of the adjustment disorder and BMI, were the reduced sample size did not allow to performed statistical contrasts. Also, the extrapolation to the general population of our findings could not be possible.
Comment. Thank you for the opportunity to revise your valuable work. I hope that my comments help to improve your paper.
Response. thank you in advance for your comments, we believe your comments really improved our work.
"Please see the attachment."

Reviewer 2 Report
Dear authors, congratulations on your research. My considerations after reading it are: In the summary, I suggest writing it without dividing by sections, better not to use the BMI but the Body Mass Index. The introduction, although brief, is concrete and clear. Perhaps it would be good to include what other studies there are in the Mexican population in this regard and what they focus on or affirm. In section 2.1 "study population" it could be specified if the information was collected with the hospital staff, or if the patients were referred once they were diagnosed. Also explain if they are recently diagnosed patients or with follow-up by the hospital. Also if the questionnaires have been carried out with the parents or in private. The instruments used are included in this section, it can be positive to make a different section with this information and also include what sociodemographic data was collected. In addition, in the description of the tests, they can include the reliability measures and confirm if they are validated in the Mexican population. Regarding the statistical analysis and the results are expressed very clearly. I especially like the results section. The discussion or conclusions may include some suggestions for the hospital and non-hospital environment that take into account the results found. These are my suggestions, receive a cordial greeting and my congratulations for the work.Author Response
Reviewer #2.
Comments and Suggestions for Authors
Comment. Dear authors, congratulations on your research. My considerations after reading it are: In the summary, I suggest writing it without dividing by sections, better not to use the BMI but the Body Mass Index.
Response. Thank you for your review, we removed the section and use Body Mass Index.
Comment. The introduction, although brief, is concrete and clear. Perhaps it would be good to include what other studies there are in the Mexican population in this regard and what they focus on or affirm.
Response. Thank you, we added a section of the studies performed in Mexican popualtion. However, in Mexican population and furthermore in adolescents, there is limited information about the prevalence of comorbid psychiatric conditions among adolescent patients with ED, where most studies had been performed in adolescents that migrate or born in the United States (doi: 10.1001/archgenpsychiatry.2011.22;doi: 10.1002/eat.20406). In Mexico, some studies had been performed in the effect of drug use comorbidity, performed only on female adolescents (doi: 10.3109/10826081003725260).
Comment. In section 2.1 "study population" it could be specified if the information was collected with the hospital staff, or if the patients were referred once they were diagnosed. Also explain if they are recently diagnosed patients or with follow-up by the hospital. Also if the questionnaires have been carried out with the parents or in private. The instruments used are included in this section, it can be positive to make a different section with this information and also include what sociodemographic data was collected. In addition, in the description of the tests, they can include the reliability measures and confirm if they are validated in the Mexican population.
Response. We added a section of the instruments.
- 3.2 Clinical measurements
Mini International Neuropsychiatric Interview for children and adolescents (MINI-kid).
The MINI-kid is a short standardized structured diagnostic interview, design for diagnostic criteria of the DSM-IV and ICD-10 for psychiatric disorders. Mini-kid disorders have shown test-retest reliability and validity comparable to other standardized diagnostic interviews (https://doi.org/10.1186/s12888-019-2121-8). In the Mexican population, it has been validated (De la Peña OF, Esquivel AG, Pérez GAJ, Palacios CL. Validación Concurrente para Trastornos Externalizados del MINI-Kid y la Entrevista Semiestructurada para Adolescentes. Rev Chil Psiquiatr Neurol Infanc Adolesc 2009;20(1):8-12).
- Questionnaire on Eating and Weight Pattern-Revised (QEWP-R).
The QEWP-R was developed by Spitzer (https://doi.org/10.1002/1098-108X(199204)11:3<191::AID-EAT2260110302>3.0.CO;2-S) to define binge-eating disorder. This version was updated to align more closely with BED criteria (doi:10.1002/eat.22372). It has been validated in different languages like Portuguese (https://doi.org/10.1590/S1516-44462005000400012) and Spanish (doi: 10.1590/S0212-16112012000200031). The QEWP-R has been reported to be a useful screening tool for BED.
Comment. Regarding the statistical analysis and the results are expressed very clearly. I especially like the results section.
Response. Thank you.
Comment. The discussion or conclusions may include some suggestions for the hospital and non-hospital environment that take into account the results found. These are my suggestions, receive a cordial greeting and my congratulations for the work.
Response. We added the following clinical relevance of our work.
- The higher BMI found on individuals with BED and ADHD is a point of clinical relevance, mainly on the effect that this increase could have in metabolic disorders, like metabolic syndrome onset (DOI: 10.1007/s11892-019-1174-x;doi: 10.1017/S1092852915000383;DOI: 10.1002/eat.22643), where this comorbidity became important not only from a mental health perspective but also from a physical perspective
- Nevertheless, the high presence of suicide risk and depression could be a delicate subject to be considered during the treatment of individuals diagnosed with ED in adolescent Mexican population. Mainly, because a high percentage of individuals with suicide ideation does not receive treatment (doi: 10.1192/bjp.bp.110.084129).
Thank you in advance for your comments, we believe your comments really improved our work.
"Please see the attachment."

Round 2
Reviewer 1 Report
The authors responded satisfactorily to all my comments.